# Assessment of Anti-Human Leukocyte Antigen (HLA)-Antibody-Dependent Humoral Response in Patients before and after Lung Transplantation

**DOI:** 10.3390/medicina58121771

**Published:** 2022-11-30

**Authors:** Anita Stanjek-Cichoracka, Marek Ochman, Elżbieta Chełmecka, Tomasz Hrapkowicz

**Affiliations:** 1Department of Biophysics, Faculty of Pharmaceutical Sciences in Sosnowiec, Medical University of Silesia in Katowice, Jedności 8, 41-200 Sosnowiec, Poland; 2Laboratory of Transplant Immunology, Silesian Centre for Heart Diseases, Szpitalna 17, 41-800 Zabrze, Poland; 3Silesian Centre for Heart Diseases, Department of Cardiac, Vascular and Endovascular Surgery and Transplantology, Medical University of Silesia in Katowice, Skłodowskiej-Curie 9, 41-800 Zabrze, Poland; 4Department of Statistics, Faculty of Pharmaceutical Sciences in Sosnowiec, Medical University of Silesia in Katowice, Ostrogórska 30, 41-200 Sosnowiec, Poland

**Keywords:** lung transplantation, anti-HLA, PRA, DSA, flow cytometry, Luminex

## Abstract

*Background and Objectives*: Testing for anti-human leukocyte antigen (HLA) antibodies both before and after transplantation is of fundamental significance for the success of lung transplantation. The aim of this study was the evaluation of anti-HLA immunization of patients before and after lung transplant who were subjected to qualification and transplantation. *Materials and Methods*: Prior to the transplantation, patients were examined for the presence of IgG class anti-HLA antibodies (anti-human leukocyte antigen), the so-called panel-reactive antibodies (PRA), using the flow cytometry method. After the transplantation, the class and specificity of anti-HLA antibodies (also IgG) were determined using Luminex. *Results*: In the group examined, the PRA results ranged from 0.1% to 66.4%. Low (30%) and average (30–80%) immunization was found in only 9.7% of the group examined. Presence of class I anti-HLA antibodies with MFI (mean fluorescence intensity) greater than 1000 was found in 42.7% of the patients examined, while class II anti-HLA antibodies were found in 38.4%. Immunization levels before and after the transplantation were compared. In 10.87% of patients, DSA antibodies (donor-specific antibodies) with MFI of over 1000 were found. *Conclusions*: It seems that it is possible to confirm the correlation between pre- and post-transplantation immunization with the use of the two presented methods of determining IgG class anti-HLA antibodies by increasing the size of the group studied and conducting a long-term observation thereof.

## 1. Introduction

Lung transplantation (LTx) is a treatment possibility that saves lives and improves quality of life in patients with end-stage respiratory failure who do not respond to any other medical or surgical interventions.

The most frequent indications for LTx are chronic obstructive pulmonary disease (COPD), interstitial lung disease (ILD), cystic fibrosis (CF) and pulmonary arterial hypertension (PAH) [1,2].

Obtaining a positive result from transplantation of parenchymatous organs, including lungs, is currently extremely difficult due to production of alloantibodies, which are directed mainly against the HLA antigens of the organ donor.

The presence of HLA antibodies is widely recognized as a barrier to solid organ transplantation. For lung transplant candidates, it has a significant negative impact on both waiting time and waiting list mortality.

The relationship between rejection of the organ transplanted through antibodies and damage to an allogeneic transplant has been described well in the case of transplantation of kidney and heart, but it has not been characterized properly so far with regard to lung transplantation.

Sensitization to HLA antigens of one’s own type takes place very easily, as these antigens are among the most sensitizing [3,4,5].

It is believed that, even after a single contact with an HLA alloantigen, memory cells retain readiness for a secondary response for many years after exposure to that antigen.

The most frequent reasons for HLA alloimmunization are: transfusions; pregnancies; other transplantations; and infections responsible for the so-called heterologous immunity, based on antigen similarity of a given pathogen and the organ that was transplanted.

Reactivation of, e.g., a virus or a repeated infection with it may cause a simultaneous immunological response of the host against both the pathogen antigens and the transplanted organ antigens.

Lung transplantation in a highly immunized recipient entails the hazard of antibody-mediated rejection (AMR) or development of the process of chronic antibody-mediated rejection of an organ. The immunological activation related to AMR in the lung includes B lymphocytes and plasma cells that create anti-HLA antibodies on the vascular endothelium in lung allotransplantation. The antigen–antibody complex formed leads to aggravation of the immune response and recruitment of cells participating in it, both through pathways dependent and not dependent on the complement, which may lead to the transplant dysfunction [6,7,8,9]. The methods that evaluate the immunological anti-HLA status of recipients include, among others, PRA-CDC (panel-reactivity antibodies complement-dependent cytotoxicity) tests conducted by means of the serological method in a test of complement-dependent cytotoxicity (CDC):-Enzyme-linked immunosorbent assay (ELISA);-Solid phase assays (SPA) employing a flow cytometer;-Solid phase assays (SPA) employing a flow fluorometer (Luminex). Employed in both the first stage of screening (where it detects the presence and class of anti-HLA IgG antibodies) and the next stage in immunized patients (a test of single antigen type, where the specificity of anti-HLA of a given class is determined), this test is mostly performed for recipients of transplanted kidney and pancreas in Poland [5,10].

The introduction of highly sensitive techniques for evaluating the specificity of anti-HLA antibodies in individual classes enabled monitoring the humoral response after lung transplantation. However, it is still not a routine action. What is of particular significance here is obviously the detection of antibodies directed against donor-specific antibodies (DSA), but it is necessary to bear in mind the patients who, after the transplantation, also produced anti-HLA antibodies other than DSA de novo [11].

The main purpose of evaluation of patient anti-HLA immunization prior to lung transplantation is determining the risk of presumptive rejection of the organ transplanted. The final decision relies on the evaluation of the obtained anti-HLA antibody results by a clinical team, leading to risk stratification, which is unique for every patient. To make these decisions, both right before the transplantation and afterwards, close interaction between the HLA laboratory and the clinical team is required [3].

According to Polish regulations and procedures, PRA-CDC is routinely determined in patients qualified for lung transplantation.

The aim of this study was the evaluation of anti-HLA-antibody-dependent humoral response of patients before and after lung transplant who were subjected to qualification and transplantation in our facility. Testing for anti-HLA antibodies before the transplantation allows for the assessment of the level of its immunization, estimating the risk of presumptive rejection and applying the right therapy before the procedure. The purpose of performing anti-HLA testing after transplantation is mainly detecting antibodies directed against the donor antigens (DSA), but also assessing whether the activation of a humoral response dependent on these antibodies directed against HLA system antigens in general has increased. This allows for an estimation of the risk of developing antibody-mediated rejection and making therapeutic attempts at stopping it. This evaluation in recipients qualified for lung transplantation was performed by determining PRA using the flow cytometry method. Meanwhile, after the transplantation, the specificity of class I and II antibodies was evaluated in patients using single antigen tests employing Luminex.

## 2. Materials and Methods

### 2.1. Patients and Samples

The examined group consisted of people who were subjected between 1 January 2018 and 30 April 2022 to the qualification and procedure of lung transplantation at the Department of Cardiac Surgery and Transplantation within the Silesian Centre for Heart Diseases in Zabrze.

The study group consisted of 152 people, of whom 89 (58.6%) were men. In that group, 9 people were below the age of 18 (age range of 13–17) and most of them were male (8 people, 88.9%). In all the analyses, underage people were not included as a separate group.

In the study group, patients with cystic fibrosis (CF) constituted the largest percentage (62 people, 40.8%), followed by those with idiopathic pulmonary fibrosis (IPF) (41 people, 27.0%), chronic obstructive pulmonary disease (COPD) (36 people, 23.7%) and primary pulmonary hypertension (PPH) (13 people, 8.5%). Meanwhile, 9 patients (3 female and 6 male) in whom the fibrosis developed after COVID-19 were included in the group with ILD.

During the course of the study conducted, 34 patients died (22.4%).

Blood samples were collected upon admission to the hospital before or after transplantation. These were collected from an antecubital vein and then placed into a BD Vacutainer. Afterwards, blood samples were immediately centrifuged at 2500× *g* for 15 min to obtain serum and stored at −20 °C, thawing only once immediately before analysis.

### 2.2. Laboratory Parameters

Prior to the transplantation, patients’ immunization was evaluated through determining the presence of anti-HLA antibodies, the so-called panel-reactive antibodies (PRA), using the flow cytometry method. These are IgG class antibodies. The examinations were carried out on an FACS ARIA sorter using FlowPRA™ Specific; One Lambda, Inc., West Hills, CA, USA.

During the first few days post-transplantation, the immunization was re-evaluated by determining the class and specificity of anti-HLA antibodies, as well as those of the IgG class, using Luminex.

HLA antibody specificities (class I and class II) were determined using the Luminex single-antigen bead assay (LABScreen™ Single Antigen HLA Class I—Combi and LABScreen™ Single Antigen HLA Class II—Group 1; One Lambda, Inc., West Hills, CA, USA).

DSA were considered positive if the mean fluorescent intensity (MFI) was >1000 for HLA class I and class II antibodies. These cut-off values were validated in the laboratory. Antibody specificities defined in Luminex were compared with the mismatched HLA antigens of the donor allograft (donor-specific antibodies, DSA). For the purposes of further analysis, the results of non-specific antibodies were also taken into account.

### 2.3. Statistical Analysis

Distribution of variables was evaluated by means of the Shapiro–Wilk test and quantile–quantile plot. The interval data were expressed as a mean value ± standard deviation in the case of a normal distribution or as a median (lower–upper quartiles) in the case of data with skewed or non-normal distribution. For comparison of data, the Student’s *t*-test or the Kruskal–Wallis one-way analysis of variance was used.

Different tests were used to determine the relationship between the qualitative characteristics. Depending on the number, these were the χ^2^ test, the χ^2^ test with Yates’ correction or the Fisher’s exact probability test. To measure the association between qualitative variables, the odds ratio (OR) with 95% confidence interval (CI) was calculated.

Statistical significance was set at a *p*-value < 0.05, and all tests were two-tailed. Statistical analysis was performed using Statistica 13.3 (TIBCO Software Inc.).

Descriptive statistics for the tables are included in the Appendix A.

## 3. Results

The 145 patients were qualified for the PRA analyses carried out. In seven patients, the material collected for PRA testing was not diagnostic and was unsuitable for further analysis. At the same time, these patients were not taken into account in further anti-HLA analysis after the transplant. In the group examined, the PRA results ranged from 0.1% to 66.4%, median (lower–upper quartile): 0.9 (0.5–2.8)%.

When determining the degree of immunization, the group examined was divided into people with negative immunization (results up to 7% PRA), which included as many as 90.3% of the patients examined; low immunization (i.e., up to 30% of PRA value), which included 8.3% of results; and medium immunization (results between 30% and 80% of PRA), which included 1.4% of the group examined.

For the purposes of further analyses, the low and medium immunization groups were joined.

The detailed characteristics of the group examined, depending on the degree of immunization based on the PRA results, are included in Table 1.

The statistically significant relationship between sex and PRA was demonstrated (χ^2^ = 6.06, *p* < 0.05). Male patients had PRA ≤ 7, while female patients had PRA > 7. In the male group the chance of obtaining PRA value of <7 is four times higher than that in the female group (OR = 4.18; 95% CI: 1.2–14.1). No statistically significant differences in age were found between groups (t = 0.203, *p* = 0.839).

No relationship was demonstrated between PRA and the disease entity (χ^2^ = 0.76, *p* = 0.857). The precise values of PRA depending on the disease entity are shown in Figure 1.

No relationship was found between death and the underlying disease (χ^2^ = 2.75, *p* = 0.431) or between death and immunization prior to transplantation (χ^2^ = 2.49, *p* = 0.114).

Three weeks after the transplantation, the immunization was evaluated once again using Luminex. For this analysis, 138 patients were qualified, since seven patients had died prior to collection of material for post-transplant anti-HLA testing.

Tacrolimus, mycophenolate1 mofetil and prednisolone were used in immunosuppression in all patients.

The reaction results are presented in the form of mean fluorescence intensity (MFI). The MFI values are proportional to anti-HLA antibody content in the serum examined.

The group examined was divided into people with negative results (MFI = 0), a subgroup with antibodies whose results were below 1000 MFI and immunized people whose MFI was above 1000. Table 2 presents the division into the above-mentioned subgroups depending on the antibody class, without differentiating into donor-specific antibodies (DSA) and non-donor-specific antibodies (non-DSA).

Table 3 includes the detailed characteristics of patients depending on MFI values in both I and II anti-HLA class.

No relationship was demonstrated between the underlying disease and immunization after the transplantation by means of increase in anti-HLA MFI (*p* = 0.718).

No relationship between death and immunization after the transplantation was demonstrated either (χ^2^ = 0.54, *p* = 0.746).

Immunization levels before and after the transplantation were compared. The result obtained borders on statistical significance (*p* = 0.080), which may suggest that enlarging the study group will lead to obtaining a statistically significant result. In 50.7% of people from the group examined, no immunization was found either before or after the transplantation, while in four people (i.e., 2.6%) low and medium immunization was found before the transplantation and high immunization after.

Class I and class II anti-HLA were summarized in the individual subgroups previously described. The results are presented in Table 4.

When summarizing the immunization class prior to the procedure (PRA), no relationship was found with class I anti-HLA (χ^2^ = 0.16, *p* = 0.069), while a weak positive correlation with class II anti-HLA was demonstrated (χ^2^ = 0.18, *p* < 0.05).

In the case of immunization after the procedure, an average positive relationship with class I anti-HLA was demonstrated (χ^2^ = 0.26, *p* = 0.01), as well as an average positive correlation with class II anti-HLA (χ^2^ = 0.27, *p* < 0.01).

In 10.87% (15) of patients, DSA antibodies with MFI of over 1000 were found, and 53.3% of those patients had class II of these antibodies. This group consisted of eight women and seven men. The largest number of people with DSA with more than 1000 MFI was in the subgroup with idiopathic lung fibrosis (40%, i.e., six people). However, it needs to be underlined that in the whole examined group there were 36 patients with DSA with MFI above 1000. This subgroup requires particular immunization evaluation control in the years after transplantation, especially in the case of their clinical condition worsening.

In the nine patients who developed fibrosis after experiencing COVID-19 infection, the PRA results ranged from 0.5% to 58%. Two patients had PRA values above 7%. After lung transplantation, four patients had MFI values >1000 both in the first and second class of anti-HLA. DSA was determined in three patients, of whom only one remains alive.

## 4. Discussion

The evaluation of the immunization degree both before and after transplantation is of key significance in terms of the criteria for proceeding with the patient, both when qualifying a patient for transplantation and after performance thereof.

Our own studies demonstrated that immunization prior to transplantation, found on the basis of the PRA results obtained using a flow cytometer, was present in 9.7% of patients and those patients had low and medium PRA values.

Our studies confirmed the statistically significant relationship between sex and PRA (*p* < 0.05) as a group are more predisposed to the formation of anti-HLA antibodies, mainly in relation to pregnancies. Our own research confirmed this relationship, as a statistically significant difference in PRA values before the transplantation depending on sex was demonstrated. Abbes et al. arrived at similar conclusions.

Low immunization in patients prior to lung transplantation was also found in the studies of Karolak et al. In the group of 121 people qualified for lung transplantation, the PRA-CDC median was 0% (min. 0; max. 53), and vPRA (virtual panel-reactive antibody) calculated according to HLA ABDR (>2000 MFI cut-off) was 8% [12,13].

When evaluating the degree of immunization in patients after lung transplantation on the basis of anti-HLA antibody specificity determinations in individual classes using Luminex, a significant increase in the number of patients in whom anti-HLA antibodies with MFI values above 1000 had been detected was found. As many as 42.7% of patients had class I anti-HLA antibodies, while 38.4% of patients had class II anti-HLA antibodies. When analyzing these results in more detail it was demonstrated that, in contrast to the relationships found before the transplantation, more men than women produced the anti-HLA antibodies with MFI values above 1000 in both classes (58% vs. 42% in class I and 55% vs. 45% in class II). However, this relationship was not statistically significant.

There are few reports regarding evaluation of non-specific anti-HLA antibodies in patients after lung transplantation. This issue has already been noticed, however, in people after kidney transplants.

According to Briggs et al., patients after kidney transplantation often have non-donor-specific HLA antibodies (non-DSA). The levels of non-DSA may fall slowly during the first month after transplantation, or may rise initially in some patients, especially during rejection with increased DSA synthesis. It may be caused by antibodies binding to common epitopes on donor-specific and non-donor-specific HLA or due to non-specific immunological reaction. Further monitoring of non-DSA levels and other immunoglobulins may lead to interesting conclusions regarding control over production of DSA.

It is also necessary to remember one other aspect, presented in the study of Rebellato et al.: high non-DSA concentrations may cause severe future difficulties when it comes to finding a compatible donor in the event of retransplantation. Obtaining information regarding non-DSA may lead to minimizing future HLA incompatibilities between the donor and recipient [14,15].

When comparing class I and II anti-HLA depending on the MFI value in the previously distinguished subgroups, a significant statistical relationship was demonstrated between these classes (*p* < 0.01).

The study group was also divided depending on the primary disease that led to the qualification and the lung transplantation. No relationship was demonstrated between PRA and the disease entity prior to the transplantation, nor between the underlying disease and immunization after the transplantation by means of increase in anti-HLA MFI.

In addition, no relationship was demonstrated between death and immunization before and after the transplantation.

Anti-HLA-antibody-dependent humoral resistance before and after the transplantation was compared. The result obtained, bordering on statistical significance (*p* = 0.080), may suggest that further studies and enlarging the study group may confirm the relationship between PRA before the transplantation and anti-HLA MFI values after the transplantation.

However, it is necessary to underline that a weak positive correlation was demonstrated between PRA values before the transplantation and the class II anti-HLA MFI values. Such a relationship was not found in the case of class I.

Our own studies prove that, in patients with positive PRA results prior to transplantation, it is also reasonable to perform a determination of specificity of these antibodies. According to Ius et al., the presence of circulating anti-HLA antibodies prior to transplantation, detected using the LABscreen tests at MFI > 1000, increases the risk of development of early DSA after the transplantation.

The studies conducted by Shah et al. and Hadjiliadis et al. demonstrated the deleterious role of complement-binding preformed DSA on graft and patient survival after lung transplantation [16,17,18].

The demonstration of the presence of DSA antibodies with MFI greater than 100 in 10.9% of patients in our own studies is very important. In class I, 53.3% had these antibodies. Zielińska et al. arrived at similar conclusions in the case of kidney transplantations. In their study group, de novo DSA antibodies were detected in 12 out of 74 anti-HLA (+) recipients [5].

According to Adei et al. and Dick et al., what may contribute to the low detectability of DSA-specific antibodies is their adsorption in the transplant tissue.

After transplantation, 26.1% of patients had DSA with MFI below 1000. This group should be encompassed by special control over specificity of anti-HLA antibodies, especially in the case of negative changes in the clinical condition of its members.

It is also known that the de novo development of donor-specific antibodies (dnDSA) results in transplantation failure. This problem is perfectly known in the case of renal transplants.

The appearance of dnDSA after transplantation is related to chronic lung allograft dysfunction and worse prognosis for the patient, which is why antibody monitoring after the transplantation is highly recommended, as it allows starting appropriate therapy as soon as possible [19,20,21].

In 2005, Girnit et al. indicated a possible impact of dnDSA antibodies taking place after lung transplantation. In the study group, patients with dnDSA had a higher acute rejection index in comparison to patients who were not creating antibodies de novo. These retrospective data demonstrated that dnDSA antibodies may cause transplantation failure. Additionally, Kauke et al. and Morell et al. demonstrated that dnDSA is related to a lower survival rate after lung transplantation.

According to Dick et al., dnDSA appeared in 20–50% of patients after lung transplantation. However, their pathophysiological impact still has not been fully learned, as in some patients with the presence of dnDSA, a rapid reduction in lung function and shortened survival time occurs, while in others it does not [21,22,23,24,25].

In the studies of Roux et al., Tikkanen et al. and Young et al., the majority of dnDSA that usually appear during the first year after transplantation are directed against HLA-DQ.

Unfortunately, such an analysis was not presented in our own studies, since in Poland DQ antigen typing in donors was not introduced until 2021.

Some interesting studies were conducted by Kumata et al., who first performed PRA evaluation in 93 Japanese patients after lung transplantation and then evaluated DSA levels in those patients with positive results. They demonstrated positive PRA values in 24.7% of patients and detected DSA in 5.4% of the study group. The positive results occurred mainly in patients with recent transplants [9,26,27,28].

The expansion of anti-HLA antibodies studies may have fundamental significance in proper therapeutic procedure, both before and after the transplantation, leading initially to lack of problems with immunological selection of donors and later to the best possible functioning of the transplanted lungs and long lifetime of the recipient.

Another important aspect of this work is considering our scheme of procedure with patients immunized before and after lung transplantation in our attempts at the unification of Polish guidelines.

## 5. Conclusions

The appearance of both DSA and non-DSA antibodies in patients after lung transplantation should lead to continuous monitoring of their anti-HLA-dependent humoral response—which precedes the clinical changes that may lead to rejection—during the later period of observing and leading these patients. It seems that it is possible to confirm the correlation between pre- and post-transplantation immunization with the use of the two presented methods of determining IgG class anti-HLA antibodies, by increasing the size of the group studied and conducting a long-term observation of that group.

## Figures and Tables

**Figure 1 medicina-58-01771-f001:**
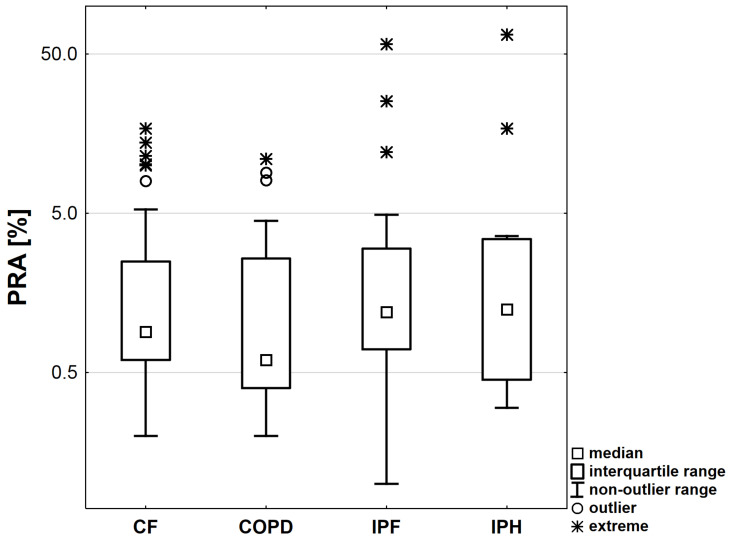
PRA (%) value depending on the disease. In the graph, the square markers represent the median, the boxes represent the interquartile range and the whiskers the range of non-outliers (o—outliers and *—extreme values).

**Table 1 medicina-58-01771-t001:** Characteristics of the study group depending on the degree of immunization based on the PRA results.

	PRA *N* = 145	
	≤7 *N* = 131	>7 *N* = 14
Sex		
Male	82 (63%)	4 (29%)
Female	49 (37%)	10 (71%)
Age (years) Mean ± SD (min–max)	38.2 ± 15.6 (10.0–70.0)	37.3 ± 13.6 (18.0–60.0)
Diagnosis		
COPD	30 (23%)	3 (21%)
IPF	36 (27%)	2 (14%)
IPH	10 (8%)	3 (21%)
CF	55 (42%)	6 (43%)

Values are n (%) or mean ± SD (min–max); COPD—chronic obstructive pulmonary disease; IPF—idiopathic pulmonary fibrosis; IPH—idiopathic pulmonary hypertension; CF—cystic fibrosis.

**Table 2 medicina-58-01771-t002:** The size of the study group depending on the MFI value in individual anti-HLA classes.

MFI Value	Class I Anti-HLA	Class II Anti-HLA
0	39 (28.3%)	41 (29.7%)
<1000	40 (29%)	44 (31.9%)
>1000	59 (42.7%)	53 (38.4%)

**Table 3 medicina-58-01771-t003:** Detailed characteristics of patients depending on MFI values in both I and II anti-HLA class.

	Anti-HLA I *N* = 138			Anti-HLA II *N* = 138		
	Negative *N* = 39	≤1000 *N* = 40	>1000 *N* = 59	Negative *N* = 41	≤1000 *N* = 44	>1000 *N* = 53
Sex						
Male	25 (64%)	21 (53%)	34 (58%)	25 (61%)	26 (59%)	29 (55%)
Female	14 (36%)	19 (47%)	25 (42%)	16 (39%)	18 (41%)	24 (45%)
Age (years) (min–max)	41.1 ± 16.0 (16–70)	35.5 ± 14.9 (10–62)	38.8 ± 15.8 (14–66)	43.0 ± 15.7 (14–70)	33.2 ± 15.8 (10–63)	39.4 ± 14.5 (13–66)
Diagnosis						
COPD	11 (28%)	11 (28%)	10 (17%)	11 (27%)	6 (14%)	15 (28%)
IPF	11 (28%)	5 (13%)	18 (31%)	11 (27%)	9 (20%)	14 (26%)
IPH	2 (5%)	4 (10%)	6 (10%)	3 (7%)	5 (11%)	4 (8%)
CF	15 (38%)	20 (50%)	25 (42%)	16 (39%)	24 (55%)	20 (38%)

Values are n (%) or mean ± SD (min–max); COPD—chronic obstructive pulmonary disease; IPF—idiopathic pulmonary fibrosis; IPH—idiopathic pulmonary hypertension; CF—cystic fibrosis.

**Table 4 medicina-58-01771-t004:** The comparison of class I and class II anti-HLA levels in individual patient subgroups.

	Anti-HLA Class II			
		0	≤1000	>1000
Anti-HLA class I	0	15	12	12
	≤1000	11	20	9
	>1000	15	12	32

A relationship was found between class I and class II of anti-HLA (χ^2^ = 15.11; *p* < 0.01).

## Data Availability

The data presented in this study are available on request from the corresponding author.

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
