# Peer review of "Assessment of Anti-Human Leukocyte Antigen (HLA)-Antibody-Dependent Humoral Response in Patients before and after Lung Transplantation"

_medicina, 2022, doi:10.3390/medicina58121771_

Round 1
Reviewer 1 Report
In this paper, Cichoracka et al. provided exciting observations regarding the presence of anti-HLA antibodies before and after transplantations. The experimental data is adequate for such studies. However, I only have some minor comments which can help readers to understand and use the data better:
1- Several single sentences can be merged as a paragraph throughout the manuscript. 2- Acquired immunity is not fast if no prior immunological memory exists. Therefore, it may take several days (or weeks) to develop an immunological response against the donor organ for patients without any alloantibodies. In addition, patients often receive immunosuppressive drugs that may further slow down this process. Therefore, authors should specify when they took blood after transplantation and any medications they used before or after transplantation if such drugs had immunosuppressive effects. This might help the conclusion for some patients if the level of alloantibodies didn't rise significantly after transplantation. Also, if there were several differences in time points when blood was collected, it should be noted and considered in their analysis. 3- The author may restructure their discussion when comparing the other studies. There are many X et al. They can reference them without mentioning their names, and if some previous studies obtain similar conclusions, sum them up.Author Response
Dear Reviewer,
Thank you for your review. I present the responses to your review.
- I tried to combine individual sentences into a single paragraph.
- The blood for testing anti-HLA levels after transplantation was collected 3 weeks after the transplant. Tacrolimus,mycophenolate mofetil and prednisolone were used in immunosuppression. (I will include this information in the manuscript)
- There were no different timepoints in the post-transplantation analysis presented.
- I have re-edited the discussion in line with the guidelines of the reviewer. I am very grateful for all the valuable comments that will surely enrich this paper and make its message clearer.KInd regards Anita Stanjek-Cichoracka
Reviewer 2 Report
Evaluation of humoral immunisation of patients before and after lung transplant
The authors aim to evaluate the humoral immunisation of patients before and after lung transplant. Prior to the transplantation, they checked for the presence of IgG class anti-HLA antibodies (anti-human 16 leukocyte antigen; panel-reactive antibodies using the flow cytometry and after the transplantation, the class and specificity of anti-HLA antibodies (also IgG) were determined using Luminex. They suggest the presence of a possible correlation between pre- and post-transplantation immunization.
However:
1. Title:
- Need to be more specific (immunization against what??)
- Or it is sensitization to HLA antigens??
2. Abstract:
- Lack the structure (background, aims, methods, results, conclusion).
- Background is needed to clarify which immunization.
3. Introduction:
- Is clear & talks about sensitization or previous exposure to HLA antigens.
- This is not immunization.
4. Methods:
- Study design is unclear.
- Inclusion & exclusion criteria are not mentioned.
- Ethical approval and Heliniski guidelines are not mentioned.
- Sample size is 152 as in methods or 145 as in results or 121 as in table 2 or 138 as in table 3???
- Patients who died, before or after sampling??
5. Discussion:
- Clear, however, more clarification of the underlying mechanisms for this sensitization is recommended
- References:
- Most of reference are old, only one in 2021, and 2 in 2020
Author Response
Dear Reviewer,
Thank you for your review.
I present the responses to your review.
- I have made changes in the title in line with the guidelines: Assessment of anti-HLA-antibody-dependent humoral response in patients before and after lung transplantation
- I have introduced the following structure in the abstract: background, aims, methods, results, conclusion.
- I have introduced corrections in the introduction and methodology sections.
- As for the inclusion and exclusion criteria, the number of transplantations carried out was 152. In 7 pre-transplant patients, the material collected for PRA testing was not diagnostic and was unsuitable for further analysis. At the same time, these patients were not taken into account in further anti-HLA analysis after the transplantation. Another 7 patients died soon after the surgical procedure and no material for anti-HLA testing was collected from them. In the end, results from 138 patients were subjected to the analysis. Table 2 includes data for 138 patients.
- I have supplemented informations about Declaration of Helsinki and the Consent of the Ethics Committee .
- I have improved the discussion
- I have supplemented the literature.
I am very grateful for all the valuable comments that will surely enrich this paper and make its message clearer. Kind regards Anita Stanjek-Cichoracka
Reviewer 3 Report
Dear authors,
I read your manuscript with interest. Lung transplantation is becoming more common year by year and the correct identification of the humoral immunisation of patients is important.
Abstract: Well written.
Introduction: Well written. All citations appropriate. Maybe considered summing some sentences into paragraphs.
Methods: -Maybe consider using the term Cystic Fibrosis instead of mucoviscidosis as the second term is not so well known.
-The ethical committe approval of the study is not mentioned. Please add it.
Results: Well presented.
Discussion: -"Our studies confirmed the statistically significant relationship between sex and PRA 230 (p<0.05)." If this applies to confirmation of known literature, please add citation.
Best regards
Author Response
Dear Reviewer,
Thank you for your review.
I present the responses to your review.
- I have supplemented the introduction.
- I have introduced the Cystic Fibrosis term in parallel.
- I have supplemented informations about Declaration of Helsinki and the Consent of the Ethics Committee .
- I have included the literature regarding the relation between the sex and PRA.
I am very grateful for all the valuable comments that will surely enrich this paper and make its message clearer. Kind regards Anita Stanjek-Cichoracka
Round 2
Reviewer 3 Report
Dear authors,
I find the final manuscript satisfactory.
Best regards.
Author Response
Dear Reviewer,
Thank you for your comments and corrections.
Kind regards
Anita Stanjek-Cichoracka